# Development of an Imaging Technique for Boron Neutron Capture Therapy

**DOI:** 10.3390/cells10082135

**Published:** 2021-08-19

**Authors:** Nobuyoshi Fukumitsu, Yoshitaka Matsumoto

**Affiliations:** 1Kobe Proton Center, Department of Radiation Oncology, Kobe 650-0047, Hyōgo, Japan; 2Department of Radiation Oncology, Clinical Medicine, Faculty of Medicine, University of Tsukuba, Tsukuba 305-8575, Ibaraki, Japan; ymatsumoto@pmrc.tsukuba.ac.jp; 3Proton Medical Research Center, University of Tsukuba Hospital, Tsukuba 305-8576, Ibaraki, Japan

**Keywords:** BNCT, ^18^FBPA, PET

## Abstract

The development of 4-^10^B-borono-2-^18^F-fluoro-L-phenylalanine (^18^FBPA) for use in positron emission tomography (PET) has contributed to the progress of boron neutron capture therapy (BNCT). ^18^FBPA has shown similar pharmacokinetics and distribution to 4-^10^B-borono-L-phenylalanine (BPA) under various conditions in many animal studies. ^18^FBPA PET is useful for treatment indication. A higher ^18^FBPA accumulation ratio of the tumor to the surrounding normal tissue (T/N ratio) indicates that a superior treatment effect is expected. In clinical settings, a T/N ratio of higher than 2.5 or 3 is often used for patient selection. Moreover, ^18^FBPA PET is useful for predicting the ^10^B concentration delivered to the tumor and surrounding normal tissues, enabling high-precision treatment planning. Precise dose prediction using ^18^FBPA PET data has greatly improved the treatment accuracy of BNCT. However, the methodology used for the data analysis of ^18^FBPA PET findings varies; thus, data should be evaluated using a consistent methodology so as to be more reliable. In addition to PET applications, the development of ^18^FBPA as a contrast agent for magnetic resonance imaging that combines gadolinium and ^10^B is also in progress.

## 1. Introduction

Imaging techniques play an important role in boron neutron capture therapy (BNCT). This chapter describes how imaging is involved in the treatment strategy of BNCT, the history of imaging techniques, and clinical assessments.

## 2. Development of ^18^FBPA PET

BNCT is a radiotherapy technique that uses the highly specific accumulation of ^10^B-containing compounds in tumor cells, with low concentrations in the surrounding normal tissues. The fission reaction produces short-range α and ^7^Li particles, which induce a high linear energy transfer and excellent anti-tumor effect with few adverse effects. The features of ^10^B-containing compounds for BNCT are high accumulation in the tumor cells, which is washed out from blood and normal tissues immediately, and few adverse effects. Currently, only two kinds of ^10^B-containing compounds have been used in clinical practice: one is sodium ^10^B-borocaptate (Na_2_B_I2_H_11_SH, or BSH), and the other is 4-^10^B-borono-L-phenylalanine (BPA). The uptake mechanism of BPA to tumor cells mainly occurs via an L-type amino acid transporter (LAT). On the other hand, BSH does not have a tumor-specific uptake system such as BPA, and tumor uptake is performed using its property of diffusion. An imaging technique for BNCT was developed to allow the measurement of BPA density. The evaluation of the tumor uptake and metabolism of BPA is very important for BNCT. In 1991, Ishiwata developed ^18^F (half-life; 109.8 min) rebelled 4-^10^B-borono-2-^18^F-fluoro-L-phenylalanine (^18^FBPA) as a positron emission tomography (PET) probe for imaging and evaluating the pharmacokinetics of BPA in vivo [1]. Various examinations using ^18^FBPA PET were reported during the latter part of the 1990s [2,3]. So far, ^18^FBPA has played a central role as an imaging modality for BNCT.

## 3. Synthesis and Basic Research of ^18^FBPA PET

^18^FBPA is synthesized by the electrophilic substitution of BPA using carrier-added ^18^F–F_2_ produced by three pathways. Figure 1a,b show the structure of BPA and ^18^FBPA. At the beginning, carrier-added ^18^F–F_2_ production was performed using the ^20^Ne(d,α)^18^F reaction and converted to ^18^F-labeled acetylhypofluorite. A cyclotron with a comparatively large amount of energy that can accelerate deuterons is necessary to complete this process. However, little radioactivity is obtained from ^18^F-F_2_ for clinical practice. Consequently, the amount of radioactivity and the molar activities of ^18^FBPA were very small: 440–1200 MBq and 20–130 MBq/µmol, respectively [1,2,4]. The synthesis was accomplished by fluorinating racemic BPA at first [1]. Over the decades, this process has been reviewed carefully to promote the stable and reproducible production of ^18^FBPA for use in clinical practice [5]. The ^18^FBPA accumulation in various kinds of tumors, such as melanoma, breast cancer, and brain tumors, has also been confirmed in animal models [6,7,8].

Grunewald and colleagues examined dynamic PET using hepatocellular carcinoma-bearing mice and ^18^FBPA and found that tumor radioactivity increased rapidly after injection and was maintained for about 60 min thereafter [9]. Some studies have been conducted on the difference and validity of tumor accumulation depending on the method of ^18^FBPA administration used. For example, Watanabe and colleagues examined the biodistributions of non-radiolabeled BPA and ^18^FBPA with the same dose given separately in two groups of squamous cell carcinoma-bearing mice. They compared the biodistribution of non-radio-labeled BPA and ^18^FBPA in different administration protocols, such as subcutaneous injection and continuous infusion. No biodistribution difference between non-radiolabeled BPA and ^18^FBPA was found in the tumor or normal tissues when BPA and ^18^FBPA were administered using the same protocol [10]. On the other hand, Grunewald and colleagues investigated different administration doses of BPA and ^18^FBPA to reproduce the clinical situation more closely [9]. They injected BPA (200 mg/kg) and ^18^FBPA (1.5 mg/kg) and observed a significant correlation between BPA and ^18^FBPA uptake in tumors and various organs, concluding that ^18^FBPA could predict the ^10^B concentration even when administered in a small amount. Wang and colleagues investigated the uptake of BPA and ^18^FBPA using glioma cell-bearing rats and found the uptake of BPA and ^18^FBPA to be similar between 0.5 and 1 h after injection [11]. Moreover, differences arising from the type of ^18^FBPA administration method used were studied. Yoshimoto and colleagues compared the pharmacokinetics of ^18^FBPA administered by bolus intravenous injection and continuous intravenous injection using various tumor models, including glioma cells, and found a correlation between the uptake of ^18^FBPA to the tumor administered by bolus and continuous injection methods [12].

## 4. Clinical Assessment of ^18^FBPA PET

Clinical studies using ^18^FBPA PET began in the late 1990s. Initial clinical studies were performed in patients with brain tumors or melanoma [13,14]. Mishima and colleagues examined ^18^FBPA PET for the treatment of malignant melanoma and found it to be a novel, highly effective method for the selective three-dimensional imaging of metastatic malignant melanomas and for the accurate determination of the ^10^B concentrations in the tumor and surrounding tissues [14]. Imahori and colleagues performed dynamic ^18^FBPA PET studies in patients with high-grade glioma. These dynamic PET studies of brain tumors revealed that ^18^FBPA accumulation in the tumor gradually increased after a bolus injection, and the value of PET activity divided by the integrated plasma activity reached a constant level at 42 min after injection. They concluded that ^18^FBPA PET could potentially provide data that would assist in the selection of patients for BNCT [13]. Imahori and colleagues also conducted research into image analysis methods. They used a 3-compartment model and rate constants (K1, k2, and k3) to perform a kinetic analysis [15] (Figure 2). The accumulation of ^18^FBPA was strongly correlated with the degree of malignancy, and the results of the dynamic PET analysis suggested that K1 (measuring the amino acid transport process) was a major factor determining the accumulation of ^18^FBPA. They concluded that the accumulation of ^18^FBPA was useful for predicting the ^10^B concentration in tumors and indicated that ^18^FBPA PET data could be of practical use for the diagnosis of malignancy and the direct prediction of the effectiveness of BNCT, which is still the basis of the BNCT treatment strategy [2].

^18^FBPA PET was utilized not only for predicting the ^10^B concentration in tumors, but also for treatment planning. Nichols and colleagues investigated the radiation dose emitting from the BPA, which was predicted from the ^18^FBPA accumulation, would improve treatment planning. They performed ^18^FBPA PET for six patients with glioblastoma (GBM) and incorporated ^10^B distribution maps, which were produced using ^18^FBPA PET images in actual treatment planning procedures. The isodose curves derived from the ^18^FBPA PET data were shown to differ both qualitatively and quantitatively from the conventional isodose curves derived from calculations based upon the assumption of a uniform uptake of the pharmaceutical in both tumor and normal brain regions. In particular, the isodose curve using the ^18^FBPA PET data had a naturally less concentric appearance and directly showed the ^10^B distribution. The clinical outcome of the patients who eventually received BNCT (five of the six patients) was compared using both sets of irradiation dose calculations. The irradiation dose which was produced using ^18^FBPA PET derived from the ^18^FBPA mapping data appeared to be more consistent with the clinical outcomes of the patients. They concluded that including ^18^FBPA PET data would likely improve treatment planning [16]. 

Takahashi and colleagues investigated the relationship between ^18^FBPA PET data and prognosis. They classified 22 patients with glioma according to the parameters obtained in a kinetic analysis of a dynamic PET scan. The median K1 value of the tumor was 0.033 mL/min and the survival period of patients whose K1 value of the tumor was higher than the medium value was 11 months. This was significantly shorter than 77 months for the patients with a K1 value below the medium value. They concluded that the K1 value, which is known to reflect the activity of amino acid transporters, was a contributing factor and suggested that treatment planning should be developed using the ^18^FBPA PET images, allowing clinicians to achieve better clinical outcomes for their patients [17]. Snimosegawa and colleagues performed repeated whole-body ^18^FBPA PET in normal volunteers to evaluate sequential changes in the ^18^FBPA concentration. They found high and moderate ^18^FBPA uptakes in the kidneys and pancreas immediately after injection. The accumulation of ^18^FBPA in other organs was very low throughout the scanning period. The peak time of the maximum concentration of ^18^FBPA was between 2.4 min (bone marrow and intestine) and 6.4 min (parotid glands). They estimated that the ^10^B concentration 50–57 min after a therapeutic dose of BPA would be approximately 7.6 parts per million (ppm) in the brain and 10.3–12.6 ppm in the salivary glands [18]. Both dynamic analysis and analysis using static image data have been used. The most representative parameter is the radioactive count ratio of tumor to normal tissue (T/N ratio), but the ratio of tumor to blood tissue (T/B ratio) is also commonly used. Isohashi and colleagues measured blood sampling data and found that the T/B ratio could be alternatively calculated by defining the region of interest as the left ventricle half an hour after ^18^FBPA injection [19]. Kankaanranta and colleagues performed a phase I/II study using 30 patients with head and neck cancer with diameters of 1.5–8.4 cm. The T/N ratio in a static image obtained between 20 and 40 min after administration was 2.5–9.0, suggesting an approximately 4-fold accumulation of BPA in the tumors compared with the corresponding normal tissues [20]. Wang and colleagues performed a phase I/II study using four patients with locally recurrent head and neck cancer with diameters of 4–9.5 cm. The T/N ratio in a static image obtained 60 min after administration was 2.3–4.5 [21]. They used a T/N ratio of >2.5 as a threshold in several of their clinical studies [22]. As ^18^FBPA PET can be used to predict the ^10^B concentration of the tumor and surrounding normal tissue, it has been included in the treatment criteria for many clinical research studies. To minimize the total radiation dose, it becomes essential that the ^10^B concentration in the tumor reaches approximately 20−35 μg/g or 10^9 10^B atoms/cell. For effective BNCT, a T/N ratio of more than 3–5 is preferable [23,24,25]. In fact, many clinical studies have set the criteria for patient selection close to this value. Aihara compared several parameters obtained from ^18^FBPA PET and evaluated their relevance to the outcomes of 10 head and neck cancer patients. They found that only the Tmin/N ratio differed between the non-complete response (CR) and the CR groups. On the other hand, the Tmax/N ratio, mean dose, minimal dose, and percentage of volume with a T/N ratio higher than 2.5 in the tumor did not differ between the groups. Therefore, they proposed using the Tmin/N ratio to improve the treatment efficiency of BNCT [26]. In a recent study in Taiwan, a total of 34 patients with malignant brain tumors in an immediately life-threatening condition who were treated by BNCT were investigated. The T/N ratio was a significant prognostic factor of cancer-specific survival, such as a 12-month survival rate with a T/N ratio < 3 (42%), 3–4 (22%), > 4 (100%), and the *p* value when the T/N ratio ≥ 4 and < 4 was 0.035, although the T/B ratio did not show a significant difference in overall, cancer-specific, and relapse-free survivals [27].

Taken together, these findings suggest that PET using ^18^FBPA is a useful and reliable treatment strategy for BNCT that is mainly used for determining the treatment indications, predicting the distribution of the ^10^B in the tumor and surrounding normal tissues, and enabling high-precision treatment planning. Figure 3a,b show brain tumor. Gadolinium enhanced lesion is found in the left parieto-occipital lobe, and abnormal accumulation is also found in ^18^FBPA PET. Figure 4a,b show tongue cancer. Huge tumor with high ^18^FBPA accumulation is seen in the floor of the right oral cavity and reaches the mandible. Figure 5a,b show melanoma. Abnormal accumulation is seen in the right neck.

## 5. Problems and Issues of ^18^FBPA PET

^18^FBPA PET has been widely used as an integral part of imaging examinations for BNCT. However, research on ^18^FBPA PET continues to have several problems that remain to be solved. Static ^18^FBPA PET images can be easily used to evaluate the ^10^B concentration, and many studies use a value relative to the value for a normal area, such as the T/N ratio. However, the area used as the “normal area”, such as muscle tissue, the same organ, the contralateral cerebellum, or the left ventricle, has not been consistent across studies. The method used to delineate the tumor has also not been standardized. Precise tumor delineation was more difficult in the 1990s and early 2000s compared with the present, since integrated PET/CT and PET/MRI systems and various types of image registration software are now available. The value of the T/N ratio naturally varies depending on how the tumor is delineated, which organ is designated as “normal”, and the size of the normal region of interest. Unfortunately, not only is the methodology not standardized, but many papers do not have detailed descriptions of these matters. Several clinical studies have progressed based on the results of T/N ratios calculated using various methods. The guidelines published in Japan in 2020 state that, for the calculation of ^10^B in tumor and normal areas, the values of different institutions can be mutually confirmed if the source of the data is clarified, enabling appropriate dose evaluations. Further improvements are needed for parameter evaluations. For ^18^FDG PET analyses, which have a long history and are used worldwide, several parameters that reflect cancer activity have been developed. Two representative parameters that have been newly developed are the metabolic tumor volume (MTV) and total lesion glycolysis (TLG). MTV and TLG are more useful for predicting prognosis than the conventional max and mean values of the standardized uptake value, and MTV and TLG have been found to reflect the tumor activity accurately in many clinical research studies examining ^18^FDG PET [28,29,30]. By reorganizing the methodology with reference to the latest research results, such as those for ^18^FDG PET, the interpretation of obtained ^18^FBPA PET data is likely to become more reliable. The calculation of the ^10^B concentration based on ^18^FBPA PET data has not yet been completely resolved. Certainly, some animal studies have revealed that the ^10^B concentration arising from a therapeutic dose can be predicted using PET and that the results for intravenous bolus and continuous injection are correlated [9,12], and the ^10^B concentration in normal tissue can be predicted using a kinetic analysis in humans [18]. However, some concerns remain regarding how accurately a small amount of ^18^FBPA accumulation can reflect the ^10^B concentration of BPA in the tumor and surrounding normal tissues when a continuous injection for 1 h or more is performed in humans. In clinical settings, the amount of neutron irradiation and the final dose distribution of the tumor and surrounding normal tissues are predicted using ^18^FBPA accumulation. 

To increase the reliability of ^18^FBPA PET and to further enhance its significance in clinical practice, a clearer demonstration that the T/N ratio of ^18^FBPA PET image can accurately forecast the distribution of ^10^B after the continuous intravenous injection of BPA is needed.

## 6. Imaging Techniques Other Than ^18^FBPA PET

Other than ^18^FBPA, some research using ^11^C-methionine (Met) has been performed. ^11^C-Met is the radiolabeled (^11^C: half-life; 20.4 min) amino acid that is most commonly used. Methionine is an essential amino acid that has an integral role in protein synthesis because it is coded for by the initiation codon. Radiolabeled methionine is a favorable and easily evaluated radiochemical product because it can not only be synthesized rapidly but also has a high radiochemical yield [31]. ^11^C-Met is an excellently sensitive method by which we can obtain a substantial amount of information on protein synthesis. Moreover, it has often been applied for delineating tumor margins, detecting tumors, and differentiating various cancers from reactive changes [32,33,34]. In 2009, Nariai and colleagues examined a comparative study of ^18^FBPA and ^11^C-Met PET in 12 patients with malignant glioma and found that the T/N ratios of ^18^FBPA and ^11^C-Met showed significant linear correlations. They suggested that ^11^C-Met is available at many clinical PET centers and that it can be used for the selection of BNCT candidates among patients with malignant glioma [35]. Yamamoto and colleagues performed a comparative study of ^18^FBPA and ^11^C-Met to evaluate BPA accumulation in a phase II BNCT study of GBM [36]. Watanabe and colleagues also performed a comparative study of ^18^FBPA and ^11^C-Met PET in seven head and neck cancer patients. They showed a strong correlation between the tumor SUVmax values of ^18^FBPA and ^11^C-Met. Moreover, they reported that ^11^C-Met PET could be used instead of ^18^FBPA PET to determine the indication patients for BNCT [37]. ^11^C-Met has a similar tumor uptake to that of ^18^FBPA; thus, ^11^C-Met could be a candidate alternative to ^18^FBPA for patient selection, especially in facilities where ^18^FBPA synthesis is not possible. However, the uptakes in some normal organs, for example the submandibular glands, heart, stomach, liver, spleen, pancreas, and bone marrow, are not always consistent [37]; thus, ^11^C-Met PET is unlikely to be a perfect alternative to ^18^FBPA PET, which can also be applied to dose distribution calculations in treatment planning.

Although few in number, some studies have applied magnetic resonance imaging (MRI). The advantage of a boron-gadolinium (Gd) compound is that it is less invasive and is widely versatile with fewer subject and facility restrictions, as well as having a high spatial resolution. Because MRI has an excellent spatial resolution for soft tissues, it is suitable for BNCT, which is often used to treat tumors in soft tissues, such as head and neck cancers and melanoma. Several agents that conjugate Gd and boron have been developed for use as contrast agents for MRI [38,39,40,41]. Although the development of these agents is still at the animal experimentation stage, these agents are expected to be applied clinically.

## 7. Conclusions

^18^FBPA PET has played an important role in treatment strategies involving BNCT. ^18^FBPA PET enables patient selection to allow a good prognosis and supports the calculation of dose distributions. The higher the ^18^FBPA accumulation ratio of a tumor and normal organs, the higher the concentration ratio of ^10^B, enabling a higher anti-tumor effect with fewer adverse effects. Many clinical studies have demonstrated this theory. Generally, the ^18^FBPA accumulation ratio of the tumor and normal organs should be 2.5 or higher. However, the method used to achieve this measurement has not been standardized, and careful interpretation is needed. In addition to conventional parameters, such as the max and mean value of the tumor, new parameters that are already used for ^18^FDG PET or the development of kinetic analyses are also expected. Moreover, the development of other agents, such as contrast media for MRI, can be expected in the future.

## Figures and Tables

**Figure 1 cells-10-02135-f001:**
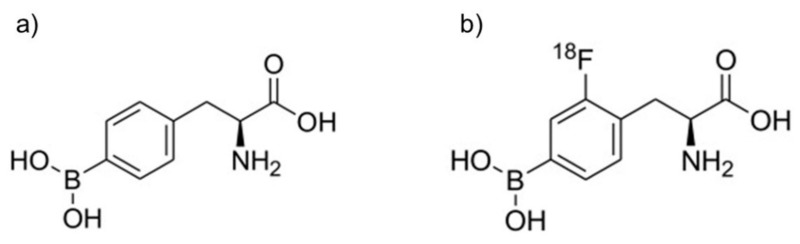
Structure of BPA and ^18^FBPA. (**a**) BPA, (**b**) ^18^FBPA.

**Figure 2 cells-10-02135-f002:**
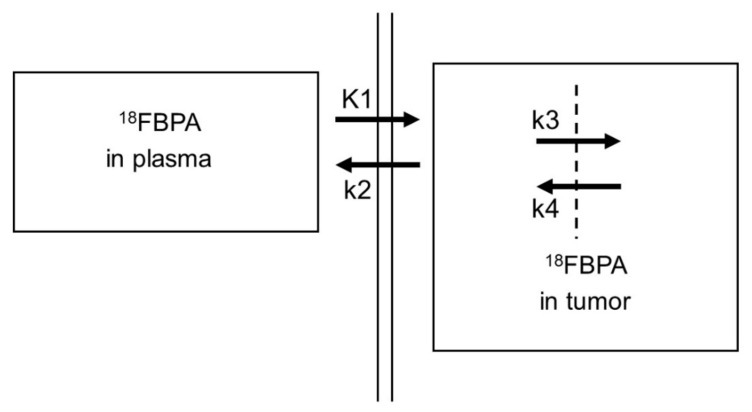
Kinetic model of ^18^FBPA to tumor cells. ^18^FBPA: 4-^10^B-borono-2-^18^F-fluoro-L-phenylalanine. K1–k4: rate constants.

**Figure 3 cells-10-02135-f003:**
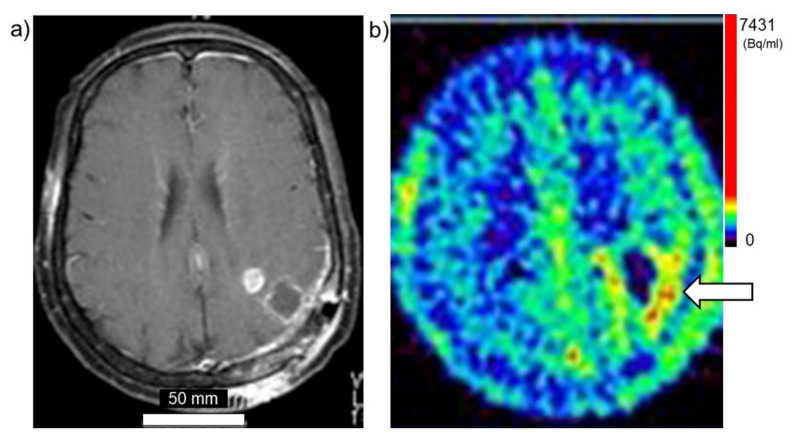
Brain tumor (glioblastoma). (**a**) MRI, (**b**) ^18^FBPA PET. Arrow presents tumor lesion. Accumulation is shown in rainbow-color scale (high in red and low in blue).

**Figure 4 cells-10-02135-f004:**
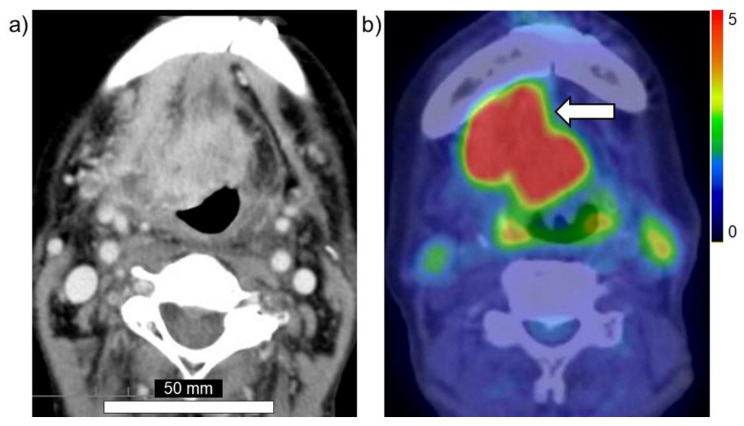
Head and neck cancer (tongue cancer). (**a**) MRI, (**b**) ^18^FBPA PET. Arrow presents tumor lesion. Accumulation is shown in rainbow-color scale (high in red and low in blue).

**Figure 5 cells-10-02135-f005:**
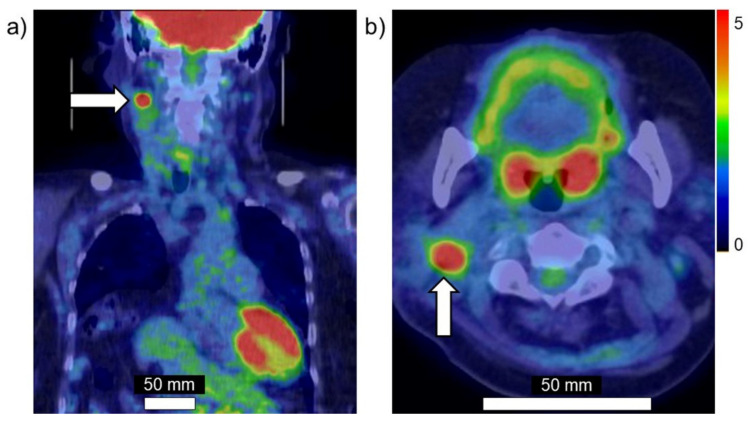
Melanoma. (**a**) Coronal image, (**b**) axial image of ^18^FBPA PET. Arrow presents tumor lesion. Accumulation is shown in rainbow-color scale (high in red and low in blue).

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
