# Peer review of "Development of an Imaging Technique for Boron Neutron Capture Therapy"

_cells, 2021, doi:10.3390/cells10082135_

Round 1

Reviewer 1 Report

I don´t understand if Figures 3-5 are new or taken from previous works. Please, clarify it. In case are not new, the images must be reference in the text and in the corresponding Figures captions. 

Author Response

Review 1

Comments and Suggestions for Authors

I don´t understand if Figures 3-5 are new or taken from previous works. Please, clarify it. In case are not new, the images must be reference in the text and in the corresponding Figures captions. 

Reply: Figure3 3-5 are new and have not been published in any journals.

Reviewer 2 Report

his is a review article for 18FBPA-PET and authors tried to recruit the latest information about the role of 18FBPA-PET for BNCT.

As authors mentioned well, 18FBPA-PET plays an important role in the treatment policy for BNCT. Without 18FBPA-PET, it’s difficult to do BNCT accurately. This in an informative article with plenty knowledge and worth to be reported. There are still some small recommendations listed as follows:

1). The first sentence in the Abstract section is mis-writing. Please check it and re-write again.

2). In the section 3. Synthesis and basic research of 18FBPA-PET, authors tried to list many researches to support the role of FBPA-PET. Wang et al has published an important article in 2004 about the role of FBPA for BNCT [J Nucl Med ;45(2):302-8]. This reference makes the research conclusion much earlier than the reference [9]. Please consider to cite this useful information and enrich the content of the article.

3). In the section 4. Clinical assessment of 18FBAP-PET, authors tried to list many clinical papers to support the use of 18FBPA-PET before BNCT. However, recently Chen et al. also published the article about the clinical use of FBPA-PET for brain tumor use in 2021 [Biology (Basel). 2021 Apr 15;10(4):334]. Please consider to cite this latest article to enrich the content of the article.

4). The authors tried to showed 3 clinical figures of 18FBPA-PET (Figure 3-5, in page 5-6). However, it lacks to mention the sources of these beautiful images. Please try to list the origins of these clinical images.

5). In the page 7 line 214, it’s neutron irradiation, not neutral irradiation. Please try to revise it.

To sum up, this is a well-written article with much important information about 18FBPA-PET. Please try to revise the recommendation stated above as soon as possible.

Author Response

Review 2

This is a review article for 18FBPA-PET and authors tried to recruit the latest information about the role of 18FBPA-PET for BNCT.

As authors mentioned well, 18FBPA-PET plays an important role in the treatment policy for BNCT. Without 18FBPA-PET, it’s difficult to do BNCT accurately. This in an informative article with plenty knowledge and worth to be reported. There are still some small recommendations listed as follows:

1). The first sentence in the Abstract section is mis-writing. Please check it and re-write again.

Reply: We corrected it.

2). In the section 3. Synthesis and basic research of 18FBPA-PET, authors tried to list many researches to support the role of FBPA-PET. Wang et al has published an important article in 2004 about the role of FBPA for BNCT [J Nucl Med ;45(2):302-8]. This reference makes the research conclusion much earlier than the reference [9]. Please consider to cite this useful information and enrich the content of the article.

Reply: We added the reference and rewrote the text.

3). In the section 4. Clinical assessment of 18FBAP-PET, authors tried to list many clinical papers to support the use of 18FBPA-PET before BNCT. However, recently Chen et al. also published the article about the clinical use of FBPA-PET for brain tumor use in 2021 [Biology (Basel). 2021 Apr 15;10(4):334]. Please consider to cite this latest article to enrich the content of the article.

Reply: We added the reference and rewrote the text.

4). The authors tried to showed 3 clinical figures of 18FBPA-PET (Figure 3-5, in page 5-6). However, it lacks to mention the sources of these beautiful images. Please try to list the origins of these clinical images.

Reply: Figure3 3-5 are new and have not been published in any journals.

5). In the page 7 line 214, it’s neutron irradiation, not neutral irradiation. Please try to revise it.

Reply: We corrected it.

To sum up, this is a well-written article with much important information about 18FBPA-PET. Please try to revise the recommendation stated above as soon as possible.

Reviewer 3 Report

This study is a review on the recent development of BNCT imaging technology using 18F BPA, etc.
I think that the latest knowledge on the development process, mechanism, clinical usefulness and limitations of 18F BPA to overcome the current limitations in terms of distribution and concentration of B has been well reviewed.
In addition to 18F BPA, reviews of new candidates under study are also briefly summarized.
There seems to be no particular problem, and if there are any recent clinical studies, it would be good to briefly introduce them.

Author Response

Review 3

This study is a review on the recent development of BNCT imaging technology using 18F BPA, etc.
I think that the latest knowledge on the development process, mechanism, clinical usefulness and limitations of 18F BPA to overcome the current limitations in terms of distribution and concentration of B has been well reviewed.
In addition to 18F BPA, reviews of new candidates under study are also briefly summarized.
There seems to be no particular problem, and if there are any recent clinical studies, it would be good to briefly introduce them.

Reply: Thank you for your comment. We added some references and rewrote the text.